# Preparation and Characterization of a Novel Swellable and Floating Gastroretentive Drug Delivery System (*sf*GRDDS) for Enhanced Oral Bioavailability of Nilotinib

**DOI:** 10.3390/pharmaceutics12020137

**Published:** 2020-02-06

**Authors:** Hong-Liang Lin, Ling-Chun Chen, Wen-Ting Cheng, Wei-Jie Cheng, Hsiu-O Ho, Ming-Thau Sheu

**Affiliations:** 1School of Pharmacy, College of Pharmacy, Kaohsiung Medical University, Kaohsiung 80708, Taiwan; hlglin@kmu.edu.tw; 2Department of Biotechnology and Pharmaceutical Technology, Yuanpei University of Medical Technology, Hsinchu 30015, Taiwan; d8801004@mail.ypu.edu.tw; 3School of Pharmacy, College of Pharmacy, Taipei Medical University, Taipei 11031, Taiwan; hamster0115@gmail.com (W.-T.C.); englave01@hotmail.com (W.-J.C.)

**Keywords:** gastroretentive drug delivery system, GRDDS, nilotinib, swelling and floating, Kollidone SR, oral bioavailability

## Abstract

Regarding compliance and minimization of side effects of nilotinib therapy, there is a medical need to have a gastroretentive drug delivery system (GRDDS) to enhance the oral bioavailability that is able to administer an optimal dose in a quaque die (QD) or daily manner. In this study, the influence on a swelling and floating (*sf*) GRDDS composed of a polymeric excipient (HPMC 90SH 100K, HEC 250HHX, or PEO 7000K) and Kollidon^®^ SR was examined. Results demonstrated that PEO 7000K/Kollidon SR (P/K) at a 7/3 ratio was determined to be a basic GRDDS formulation with optimal swelling and floating abilities. MCC PH102 or HPC_sssl,SFP_ was further added at a 50% content to this basic formulation to increase the tablet hardness and release all of the drug within 24 h. Also, the caplet form and capsule form containing the same formulation demonstrated higher hardness for the former and enhanced floating ability for the latter. A pharmacokinetic study on rabbits with pH values in stomach and intestine similar to human confirmed that the enhanced oral bioavailability ranged from 2.65–8.39-fold with respect to Tasigna, a commercially available form of nilotinib. In conclusion, the multiple of enhancement of the oral bioavailability of nilotinib with *sf*GRDDS could offer a pharmacokinetic profile with therapeutic effectiveness for the QD administration of a reasonable dose of nilotinib, thereby increasing compliance and minimizing side effects.

## 1. Introduction

Nilotinib with the brand name of Tasigna was approved by the United States (US) Food and Drug Administration (FDA) for the treatment of chronic phase and accelerated phase Philadelphia chromosome-positive chronic myelogenous leukemia (CML). The usual dosage is 400 mg given twice daily, and it should be administered 1 h before a meal or 2 h after a meal to avoid an increase in nilotinib plasma concentration, which causes toxicity and side effects [1]. With once-daily administration, steady-state nilotinib exposure was linear in the dose range of 50–200 mg/day and was dose-dependent, with less than dose-proportional increases in systemic exposure at dose levels of >400 mg/day, possibly because of limited solubility in gastric acid or saturation of its uptake. Upon twice-daily administration, there was no relevant increase in exposure to nilotinib when the dose was increased from 400 to 600 mg/day [2]. This might have also been due to the maximal solubility in gastric acid being achieved in this dose range. At a dose of 800 mg/day, exposure to nilotinib following 400 mg twice daily was about 35% higher than after 800 mg given once daily, while the increase in exposure to nilotinib between the first dose and steady state was approximately two-fold for once-daily dosing and 3.8-fold for twice-daily dosing, both possibly because of the total solubility in gastric acid with two divided doses being higher than that for a single dose, leading to enhanced bioavailability [2,3].

The inter-individual variability of nilotinib exposure is relatively high (with a coefficient of variation (CV) of 32–64%), which can be partly explained by its solubility-limited absorption and low bioavailability [2,4]. No differences in the pharmacokinetics of nilotinib were observed between patients with CML and patients with a gastrointestinal stromal tumor (GIST) [1,5,6,7], but nilotinib absorption may decrease by approximately 48% or 22% in patients with GIST with either a total or partial gastrectomy, respectively [8,9,10]. This demonstrates that dissolution of nilotinib in gastric acid is the main factor determining its oral bioavailability. Furthermore, in 10 patients with CML, the mean area under the curve (AUC) of nilotinib increased by 50% when the drug was administered with a high-fat meal [2]. This was probably a result of its increased soluble fraction in gastric acid that was secreted by the presence of foods leading to enhanced bioavailability [11]. Additionally, when co-administered with esomeprazole, nilotinib C_max_ and AUC_0-∞_ decreased by 27% and 34%, respectively. These results overall highlight that the extent of exposure to gastric acid in the stomach principally determines the rate and extent of nilotinib absorption [12].

Since nilotinib has the effect of prolonging atrial repolarization, and this effect is directly proportional to the plasma concentration, there is concern with the toxicity problem caused by food or high-fat foods, leading to increased plasma concentrations and bioavailability [13,14]. The results of the study published by Saglio et al. further suggested that nilotinib has great potential to replace imatinib and become a first-line treatment for patients with chronic phase (Ph+) CML. However, this study also showed that nilotinib needs to be taken at 600–800 mg in one dose per day to achieve a better effect. Not only does such a high dose increase the clinical burden of the patient, but the twice-a-day model of administration also slightly affects patient compliance. If a single formulation of a pharmaceutical prescription could be developed to improve the bioavailability of nilotinib and match the dissolution mode of quaque die (QD) or daily administration, the clinical efficacy of nilotinib would certainly have great economic benefits [15].

To maximize exposure of nilotinib to gastric acid in the stomach, thereby enhancing oral bioavailability and matching the dissolution of QD or daily administration, a gastroretentive drug delivery system (GRDDS) would be the optimal choice, since it is mainly designed to allow the drug to stay in the stomach for a period of time to increase exposure to the acidic environment of the stomach, thereby increasing the bioavailability of the drug. On the other hand, nilotinib delivered by a GRDDS would increase the time the drug stays at the site of action in the stomach, leading to improved therapeutic efficacy of GIST patients [4,16]. There are four types of GRDDSs, high-density systems [17], bioadhesive (mucoadhesive) systems [18,19], floating systems [20], and swellable types [19]. Swellable-type GRDDSs can be formulated with commonly used hydrogel excipients, such as polyvinyl alcohol (PVA), polyethylene oxide (PEO), hydroxypropyl methylcellulose (HPMC), hydroxyethyl cellulose (HEC), and sodium carboxymethyl cellulose (NaCMC) [19], manufactured via direct compression in a tablet dosage form, which can expand to a certain size in a short time to avoid passing through the pylorus valve of the stomach, thus extending the gastroretention time. On the other hand, floating GRDDSs can be designed with the use of Kollicoat^®^ SR, which can be compressed at a low tableting force to have high tablet hardness, ensuring a low initial density of tablets for floating resulting in prolonged gastric retention [21,22]. GRDDSs so designed with swelling and floating abilities possess four characteristics: (1) the dosage form is small enough to be swallowed; (2) taken orally, the dosage form remains floating after being co-administrated with water to prevent gastric emptying; (3) after the dosage form reaches the stomach, it expands to a certain size in a short time to avoid passing through the pylorus valve of the stomach; (4) when the dosage form is no longer required to be gastroretentive, it can be reduced to a size which can be excreted or be decomposed in the body [23,24]. Therefore, novel swellable and floatable (*sf*)GRDDSs in tablet and capsule forms for enhanced oral bioavailability of nilotinib were prepared and characterized in this study. The in vivo oral bioavailability of a nilotinib-loaded *sf*GRDDS was examined to demonstrate the enhanced oral bioavailability to reach a pharmacokinetic profile able to offer therapeutic effectiveness by QD administration of nilotinib.

## 2. Materials and Methods

### 2.1. Materials

Kollidon^®^ SR composed of 80% *w*/*w* polyvinyl acetate (polyvinyl acetate, PVAc) and 19% *w*/*w* povidone (PVP) was supplied by BASF (Ludwigshafen, Germany). Hydroxyethyl cellulose 250HHX (with a viscosity of 3400–5000 cP, an estimated molecular weight of 1600 kDa, designated hydroxyethyl cellulose (HEC) 250HHX) was of pharmaceutical grade and was supplied by Hercules (Wilmington, VA, USA). Hydroxypropyl methylcellulose (HPMC) 90SH 100K and polyethylene oxide 7000 k (PEO 7000K) were obtained from Dow Chemical (Midland, MI, USA). Microcrystalline cellulose (MCC) PH102 was obtained from Wei Ming (Taipei, Taiwan). Hydroxypropyl cellulose (HPC_ssl,SFP_) and colloidal silicon dioxide 200 (Aerosil 200) were respectively provided by Nippon Soda (Tokyo, Japan) and Evonik Resource Efficiency (Hanau-Wolfgang, Germany). All excipients used in this study were pharmaceutical grade.

### 2.2. Preparation of Gastroretentive Tablets and Capsules

The formulation study selected three swellable and expandable polymers (PEO7000K, HPMC 90SH 100K, and HEX250HHX) with Kollidon^®^SR (BASF), which was the main substance controlling drug release, and they were prepared via a direct compression method. In addition, microcrystalline cellulose (MCCPH102; Wei Ming) and hydroxypropyl cellulose (HPCssl SFP), which were used as dissolution-enhancing agents, were incorporated in the formulation, where ratios of polymers to Kollidon^®^ SR of 10:0, 7:3, 5:5, 3:7, and 0:10 were used. All polymeric materials and excipients were firstly passed through a No. 40 mesh and mixed in a plastic bag for 3 min. Nilotinib, which was separately mixed with Aerosil^®^ 200 and sieved via mesh No. 40, was then added to the above mixture and was further mixed in a plastic bag for another 3 min, and 1% magnesium stearate (Merck, Darmstadt, Germany) was subsequently added as a lubricant with mixing for an appropriate time. Tablets were prepared by weight into a 12-mm-diameter die with the nilotinib content equivalent to 150 mg/tablet and compressed with 0.5 or 1.0 tons of force using a tablet press (Carver Laboratory Press Model C, Carver, Wabash, IN, USA). For the capsule dosage form, powder was manually filled into No. 0 capsules. Finally, the water-swelling ability of the optimized formulations was evaluated, and in vitro dissolution tests were conducted.

### 2.3. Physical Characterization of Tablet Formulations

The size, diameter, and thickness of nilotinib tablets (units: mm) was evaluated in triplicate (*n* = 3) using vernier calipers. Three tablets of each formulation were randomly selected and used to measure the hardness of the tablets (PTB-311; Pharma Test, Hainburg, Germany). Swelling studies were conducted using the Vankel Dissolution Apparatus (VK7020S, Varian, Palo Alto, CA, USA). No rotation speeds were applied. Pre-weighed tablets were immersed in 900 mL of medium (simulated gastric solution, 0.1 N HCl) and maintained for 8 h at 37.0 ± 0.5 °C. At predetermined time intervals (1, 3, 9, and 24 h), swollen tablets were removed from the solution, immediately wiped with a paper towel to remove surface droplets, and weighed. The swelling index (Sw) was calculated according to the following equation: swelling index (Sw) = W_t_ − W_0_/W_t_, where W_0_ is the initial weight of the dry tablet and W_t_ is the weight of the swollen tablet at time t. Data are presented as the mean ± standard deviation (SD) of three samples per formulation.

### 2.4. Dissolution Test

Dissolution tests were conducted in triplicate for all formulations by the apparatus II method (USP XXIX) (VK7020, Vankel, UK). All release studies were performed at 100 rpm in 900 mL of simulated gastric solution, 0.1 N HCl, at 37.0 ± 0.5 °C. Five-milliliter samples were withdrawn at predetermined intervals (0, 0.5, 1, 1.5, 2, 4, 6, 8, 10, 12, and 24 h), and were refilled with the same volume of fresh dissolution medium. The drug concentrations in the withdrawn samples at each time point were analyzed using a high-performance liquid chromatographic (HPLC) method after being filtered via a 0.22-µm filter, and appropriate dilution was performed as needed.

An HPLC method was developed to estimate nilotinib in bulk, dosage forms, and dissolution media. The method was employed on an Atlantis^®^ T3 C18 column (4.6 mm × 250 mm, 5 µm, Waters, Milford, MA, USA), and acetonitrile/water = 600:400 (*v*/*v*) was used as the mobile phase at a flow rate of 1 mL/min. The selected UV detection wavelength was 254 nm. The column temperature was set to 40 °C, and the sample injection volume was 10 μL. The HPLC method was validated by a standard curve in the concentration range of 0.5–30 µg/mL.

### 2.5. Animal Study of Oral Bioavailability

#### 2.5.1. Animal Dosing

Six male white New Zealand rabbits were selected since its pH values in stomach and intestine are both similar to human [25], and they were obtained from the Animal Center of Taipei Medical University. The rabbits, each weighing around 3.0–4.0 kg, were individually housed with free access to food and water. All experiments were approved by the animal ethics committee of Taipei Medical University (LAC-2015-0108, 15 December 2015). Before administration, each rabbit was starved for 24 h with access to drinking water and was then given a single dose, followed by a 10-day washout period.

In total, six rabbits were randomized into two groups, and each animal was given a single dose of 150 mg/cap (Tasigna) or a 15 × 4 mm gastroretentive tablet containing 150 mg of nilotinib (or a gastroretentive capsule (No. 0 capsule)). Blood was collected from each rabbit through the ear vein at 0, 1, 2, 4, 6, 8, 10, 12, 16, 24, 30, 36, 48, and 72 h after drug administration. The blood collection syringes were wetted with 100 IU/mL heparin saline. Blood samples were immediately placed in a micro-tube, shaken up and down for mixing, and centrifuged at 3000 rpm for 10 min at 4 °C; then, the upper layer was aspirated. The plasma of each sample was dispensed into microcentrifuge tubes and frozen at −80 °C until being assayed.

#### 2.5.2. Liquid Chromatography Tandem Mass Spectroscopy (LC-MSMS) Analysis of Nilotinib Plasma Concentrations

An ultra-performance LC (UPLC) analysis was conducted with an ACQUITY UPLC system, Xevo TQ MS system (Waters), and the ionization mode was the electrospray free positive ion mode. The analytical conditions are shown in Table 1 and Table 2. Methyl *tert*-butyl ether (MTBE; 1400 μL) was added to 20 μL of the imatinib (IS, 5 µg/mL) working solution and 200 μL of nilotinib plasma samples. After being thoroughly vortex-mixed for 10 min, the mixture was centrifuged at 8000 rpm and 4 °C for 15 min. The upper organic phase of the extract was transferred to another clean tube and evaporated to dryness using nitrogen. Afterward, the residue was dissolved in 200 μL of 0.1% formic acid/acetonitrile (9:1) and centrifuged and vortexed for 3 min. The clear supernatant was injected into the column for analysis.

#### 2.5.3. Statistical Analysis of the Pharmacokinetic Study

Measured values of the experimental data are expressed as the mean ± standard deviation (SD), and the relationship between the concentration of each drug in the blood sample and time, and the pharmacokinetic parameters were calculated in the non-compartment mode using WinNonlin 6.3 software (Pharsight^®^, Princeton, NJ, USA), including the maximal (or peak) plasma concentration of the drug (C_max_), time to the maximum plasma concentration of the drug (T_max_), the area under the plasma drug concentration–time curve (AUC), the elimination rate constant (K_el_), and the drug half-life (T_1/2_). The equation below was used to calculate the relative bioavailability. All values are presented as mean ± standard error (SE).
$$F_{rel}=100\times \frac{AUC_{A}/Dose_{A}}{AUC_{B}/Dose_{B}}$$

### 2.6. Statistical Analysis

All means of physical data are presented with their SD as the mean ± SD. All means of the in vitro and in vivo studies are reported with their standard error of the mean (SEM). An analysis of variance (ANOVA) was conducted. A value of *p* < 0.05 was accepted as statistically significant.

## 3. Results and Discussion

Hydrophilic polymer matrix systems are widely used in oral controlled-release dosage forms because of the flexibility of the materials, which often provides the desired drug release, reasonable economics, and acceptance by patients. Hydrophilic polymers that possess gelling properties in water are also widely used in formulating swellable GRDDSs with sustained-release characteristics. Kollidon SR can form tablets with a light density at lower compression forces, and those tablets float in fluids. Our study compared the influence on the swelling and floating caused by water absorption with hydrophilic polymeric excipients including HPMC 90SH 100K, HEC 250HHX, and PEO7000K in the presence of various amounts of Kollidon^®^ SR. Kollidon^®^ SR is mainly composed of 80% *w*/*w* PVAc and 19% *w*/*w* povidone (PVP). Kollidon^®^ SR is not sensitive to pH in the environment. PVAc is a material with high plasticity; thus, it can form matrix tablets even at low tableting pressures. When a tablet is orally administrated and exposed to gastric juice or intestinal fluid, the water-soluble povidone dissolves slowly, and pores in the matrix tablet allow the main component drug to slowly be released into the environment. Kollidon^®^ SR is also a non-ionic substance; thus, there is no bonding or chemical influence with substances of which the tablet is constructed. The hydrophilic polymeric excipient was mixed with Kollidon^®^ SR at ratios of 10:0, 7:3, 5:5, 3:7, and 0:10 (*w*/*w*), and each formulation had a diameter of 15 mm and a weight of 400 mg. Results of swelling and floating at predetermined times for those tablets are shown in Table 3 and Table 4.

Results in Table 3 and Table 4 demonstrate that, regardless of which of the three polymers, HPMC 90SH 100K, HEC 250HHX, and PEO7000K, was used, none of them could float in pH 1.2 simulated gastric acid when 1.0 ton of compaction pressure for tableting was used. However, with a tableting pressure of 0.5 tons, each of them could float, and the swelling extent as indicated by the change in tablet diameter was not inferior to that prepared at a tableting pressure of 1 ton. Among the three polymeric excipients examined, PEO7000K was observed to have the best swelling effect for tablets prepared at both compression forces of 0.5 and 1 ton. At 1 h, only the PEO7000Kalone (10:0) and PEO7000K/Kollidon SR (P/K = 7/3) group could swell to 15 mm in diameter, whereas the HPMC 90SH 100K (10/0, 7/3, and 5/5) and HEC 250HHX (10:0, 7/3, 5/5, 3/7, and 0/10) groups could only swell to <15 mm. A diameter of >15 mm might be expected to avoid passing through the gastric pylorus valve. Furthermore, as shown in Figure 1, a higher proportion of hydrophilic excipients led to a greater expansion of the structure. The diameter of the tablet could be increased to 15 mm or more and was observed to not have dissolved by simulated gastric acid at 24 h. However, the PEO7000K group had the best swelling effect, which reached nearly 25 mm in diameter by 24 h.

Furthermore, according to the results shown in Table 3, PEO7000K was found to have good water absorption and swelling effects; thus, it was selected as the hydrophilic excipient with Kollidon^®^ SR for a gastric retention formulation. When the content of PEO7000K was higher, the swelling effect was better. When the formulation was composed of PEO7000K alone, the swelling effect was the best, but this formulation could not be suspended, which increased the chance of gastric emptying. Therefore, P/K was finally selected at a 7/3 ratio as a basic formulation of the GRDDS, and drug release profiles were further examined after incorporation of the drug.

Firstly, the drug mixed with excipient (P/K = 7/3) at different proportions (drug/excipient ratio, D/E) was formulated to observe the effect of the added amount of excipient on drug release. Drug release with 5% Aerosil 200 added at a fixed drug to P/K ratio of 1/4 was also compared. Results shown in Table 5 and Figure 2A reveal that a higher added amount of excipient led to a slower drug release rate. When the D/E ratio was equal to 1/4 (A1 formulation), drug release reached only 38% in 24 h, whereas it was 95% in 24 h for D/E ratio equivalent to 1/2 (formulation A3). However, the tablet hardness for the latter was much lower than that for the former, which was unfavorable for clinical application. Therefore, Aerosil 200 was added to formulation A1 to help provoke a wicking effect of water into the tablet to rapidly induce a widespread swelling effect in the tablet. It turned out that formulation A3 with 5% Aerosil 200 in formulation A1 at the same D/E ratio of 1/4 exhibited a three-fold increased drug release rate at 24 h. Nevertheless, in the pharmaceutical industry, the addition of Aerosil 200 to formulations is usually at 0.5–1%; thus, other suitable dissolution-enhancing agents must be found to increase the drug release rate and increase the hardness of the tablet.

Figure 3A shows that 100% of Tasigna (the brand name of nilotinib), composed of 150 mg of nilotinib with Pluronic F68 as the main excipient in the capsule dosage form, was released in 1 h. The instantaneous release of nilotinib was observed. The influence on the drug release rate and tablet hardness of utilizing MCC102 as a dissolution-enhancing agent and improving the tablet hardness at various added amounts was compared. Table 6 lists formulations which incorporated MCC102 at different ratios of drug to excipient (P/K = 7/3). M1 and M2 represent the weight ratio of drug to excipient (P/K = 7/3) of 1:0.5, and M1 was formulated with the weight amount of MCC102 equivalent to 50% of the weight of P/K, while M2 had an MCC102 equivalent to 100% of the P/K weight. Dissolution results showed that drug release rates of the two groups were similar, but the hardness of M2 was higher than that of M1. The weight ratio of drug to excipient (P/K = 7/3) in M3–M6 was 1:0.5, and various amounts of MCC102 were added. As the amount of MCC102 increased, the hardness of the tablet also increased from 1.63 to 3.9 kPa. Results of the drug release profile for M5 and M6 (with MCC102 contents of 100% and 200%, respectively) as shown in Figure 2B demonstrate that the release rate at 4 h was faster than that for M3 and M4, in which no MCC102 and 50% MCC102 were respectively added, whereas the release rate at 6 h became faster instead for those groups with a lower amount of MCC102 added. At 12 h, the release rates of all groups were close to 100%. The reason for the faster release of MCC102 in the first group was presumed to be the presence of a higher content of MCC102, which caused the tablet to initially swell and expand at a faster rate leading to faster release of the drug. However, after swelling with water, the insoluble MCC102 became resistant to outward diffusion of the drug located internally, resulting in a reduction in the release rate after 6 h. Thus, the formulation with a 50% content of the dissolution enhancing agent (MCC102) was finally selected to increase the tablet hardness and release the drug within 24 h.

In order to make swallowing tablets with higher drug contents easier, the stamping die was altered into a long strip shape (15 mm in diameter) as a caplet. The formulations are shown in Table 6. F1 and F2 had weight ratios of drug to excipients (P/K = 7/3) of 1:0.5 and 1:1, and MCC102 equivalent to 50% of the P/K weight was added to the formulations. In F3 and F4, weight ratios of drug to excipients (P/K = 7/3) were 1:0.5 and 1:1, and HPC equivalent to 50% of the P/K weight was added. The so-obtained caplets all had higher hardness than the corresponding respective tablets. This might have been due to the fact that, with a smaller force area, it received a larger pressing force, and the structure also become more compact. Nevertheless, the caplet group did not float after absorbing water.

In Figure 3A, the drug release profile for Tasigna (the brand name of nilotinib) composed of 150 mg of nilotinib with Pluronic F68 as the main excipient in the capsule dosage form was demonstrated to be 100% released in 1 h. The instantaneous release of nilotinib was observed. In the MCC102 group (F1 and F2), a greater proportion of the swellable/floating excipient (P/K) led to a slower drug release. However, the HPC_ssl,SFP_ groups (F3 and F4) were not affected by the total amount of excipient (P/K) added to the formulation, and results showed that release rates for F3 and F4 were similar. The reason might be that HPC_ssl,SFP_ is a super-fine powder, which possesses excellent permeability to water. Therefore, even if the added amount of P/K increased, it still had no effect on the rate at which water entered the tablet, resulting in the rate of drug dissolution remaining the same. Between the F1 and F3 groups, it was observed that the release rate of the first 6 h for F1 was faster than that for F3. This was probably because, in the group using MCC102 as the dissolution-enhancing agent, it quickly dissolved from the periphery of the tablet, and the release rate at the previous time point was faster. Due to the compact structure of the caplet, we observed that none of the formulations in caplet form floated during the dissolution test. Therefore, it was assumed that the time period of gastric retention might be significantly prolonged by filling the same formulation as that for caplets into No. 0 capsules to reduce the density, allowing the so-obtained capsules to float.

Table 6 reveals the formulations designed for the capsule-type GRDDSs. C1–C3 were composed of excipient (P/K = 7/3) at D/E ratios of 1:0.5, 1:0.75, and 1:1, respectively, and MCC102, with an added weight equivalent to 50% of the P/K weight. The design of C4–C6 was similar to that of C1–C3, but MCC102 was replaced with HPC. Figure 3B displays drug release rate profiles of C1–C6. Regardless of whether the group contained HPC or MCC102, the drug release rate was the fastest for a D/E ratio of 1:0.5. When the D/E ratio increased to 1:0.75 and 1:1, the amount of excipient did not affect the dissolution rate, and dissolution rates of HPC formulations C4–C6 were faster and reached 100% release in 12 h, but it took 24 h to reach 100% release for MCC102 formulations C1–C3. The difference between the caplet and capsule forms was that the capsule formulations floated, and the release rate might be expected to only be affected when the P/K content was equivalent to 50% of the drug. When the content of the swelling agent was low, the drug was released as soon as the capsule disintegrated; however, when the content of swelling excipients was high enough, it formed a colloid after water absorption, which was similar to the structure of the caplet, and did not affect the dissolution rate. When the content of the swelling excipient was more than 75% that of the drug, the release rates of capsules and caplets were similar. In order to understand the effects of gastroretentive ability on oral pharmacokinetics of nilotinib, several groups of gastroretentive formulations (F1, F2, and C1–C6) with sufficient strength, floating ability, good swelling properties, and different release rates were further examined in a rabbit model.

The drug plasma concentration versus time profile of oral administration of 150 mg of Tasigna^®^ at a single dose in white rabbits is shown in Figure 4A, and oral administration of Tasigna^®^ reached the highest plasma concentration around the second hour, followed by a drug concentration decline as time went by, while the drug was not detected in the blood after 48 h. The calculated pharmacokinetic parameters are shown in Table 7. The time (T_max_) for the drug to reach the highest blood concentration was 2.66 ± 1.52 h, the highest concentration of the drug (C_max_) was 614 ± 363 ng/mL, and the area under the drug curve (AUC) was 2703 ± 1219 ng·h/mL.

The drug plasma concentration versus time profiles for oral administration of formulations F1 and F2 containing 150 mg nilotinib as a single dose in rabbits are also displayed in Figure 4A. It shows that oral administration of F1 resulted in the highest plasma concentration reached at around 15 h, followed by a declining drug concentration as time went by, and the obtained pharmacokinetic parameters are shown in Table 7. The time to the highest blood concentration of the drug (T_max_) was 15.5 ± 10.11 h, the highest drug concentration (C_max_) was 795 ± 311 ng/mL, and the area under the drug curve (AUC) was 16,728 ± 620 ng·h/mL. The orally administered F2 formulation was retained in the stomach for a longer time than F1, due to the higher content of swelling excipients; thus, the time to reach the highest blood concentration of the drug (T_max_) was extended to 24 h, the highest concentration of drug (C_max_) was 869 ng/mL, and the area under drug curve (AUC) was 19,974 ng·h/mL. The bioavailability of F1 and F2 was 6–7-fold higher compared to that for Tasigna^®^. This was mainly attributed to Tasigna^®^ being encapsulated in a capsule, leading to all nilotinib content being released instantly after quick dissolution of the capsule shell to expose its content to simulated gastric fluid (SGF). Since nearly 100% of the drug was dispersed instantly into the stomach, a smaller portion of nilotinib might have been transformed into the salt form in the stomach, leading to only this smaller portion of dissolved nilotinib being available for absorption in the major absorption site of the intestines as the dissolved nilotinib was emptied into the intestines. Additionally, as soon as the stomach emptied the drug load in the stomach into the intestines, no more nilotinib was dissolved in the intestine due to its neutral to slightly alkaline pH, thus leading to further retardation of the oral bioavailability with an increasing undissolved portion of nilotinib not being absorbed in the intestines. Nevertheless, by increasing the retention time of the drug in the stomach by the design of the GRDDS formulation, the sustained release of nilotinib from the GRDDS would be expected to gradually transform all of the nilotinib content into the salt form to dissolve in the stomach before emptying into the intestines, thus leading to enhancement of its oral bioavailability.

The drug plasma concentration versus time profiles and the obtained pharmacokinetic parameters for oral administration of formulations C1–C6 in a single dose are shown in Figure 4B and Table 7, respectively. In the C1–C3 groups using MCC102 as the dissolution-enhancing agent and hardness enhancer, a higher D/E ratio led to a greater increase in bioavailability, which increased by 2.65-, 4.73-, and 7.52-fold, respectively. Comparing the same formulations with different dosage forms, C1 and F1, although there was no difference in the in vitro dissolution profile, the in vivo pharmacokinetics were observed to differ between the capsule and the tablet. Drug in the swellable tablet could only be released by swelling or expansion to loosen the diffusion resistance inward from the outer surface, whereas drug in the capsule was available for dissolution as soon as the swellable excipient absorbed water to became a gel form. As a result, the initial release rate for the capsule was expected to be faster than that for the tablet, and the extent of drug release from the capsule form was also expected to be greater than that for the tablet. Therefore, C1 reached the T_max_ time earlier than F1, with values of 9.67 and 15.5 h, respectively. Alternatively, the extent of drug release was lower from the tablet form than from the capsular form, leading to increased bioavailability of F1 to a greater extent than for C1. When the D/E ratio was equal to 1.0, which occurred with C3 and F2 having the same formulation but different dosage forms, F2 reached T_max_ more slowly and the C_max_ concentration was lower, but C3 and F2 increased the relative bioavailability to 7.52- and 7.4-fold, respectively. It was observed that the drug bioavailability at high excipient levels showed no difference between the tablet and capsule. In the C4–C6 groups using HPC_ssl,SFP_ as a dissolution-enhancing agent and hardness enhancer, the pharmacokinetic profiles and their pharmacokinetic parameters are shown in Figure 4B and Table 7, respectively. Results illustrated that C_max_ values of these three groups were almost 1000 ng/mL, and average T_max_ values were around 12–20 h, which were delayed compared to each corresponding MCC102 group (7.0–9.7 h). However, there was no correlation between the relative oral bioavailability enhancement and the D/E ratio. Nonetheless, the enhancement multiples of oral bioavailability for C4–C6 were higher than those for each corresponding formulation of C1–C3.

## 4. Conclusions

This study combined swellable material of PEO7000K with floatable material of Kollidon^®^ SR to successfully produce an *sf*GRDDS formulation for retaining nilotinib in the stomach for a desirable period of time. Further inclusion of MCC (PH102) or HPC (ssl, SFP) as a dissolution-enhancing agent in the above *sf*GRDDS formulation could release the drug at a rate optimally sufficient to gradually convert all of the nilotinib into the salt form available for dissolution in the stomach, being maintained in the dissolved state for intestinal absorption after emptying into the intestines, which led to enhanced oral bioavailability. The drug release profiles and resulting plasma drug concentrations of the so-designed *sf*GRDDS formulations could be controlled by different D/E ratios either in tablet or capsule dosage forms. Overall, this study demonstrated that the multiples of enhancement of the oral bioavailability for nilotinib formulated with *sf*GRDDS could reach a pharmacokinetic profile that was able to offer therapeutic effectiveness by QD administration of nilotinib at a reasonable dose while reducing the number of doses taken, thereby increasing patient compliance and minimizing side effects. However, the lack of stability testing of these formulations should be considered as a potential limitation.

## Figures and Tables

**Figure 1 pharmaceutics-12-00137-f001:**
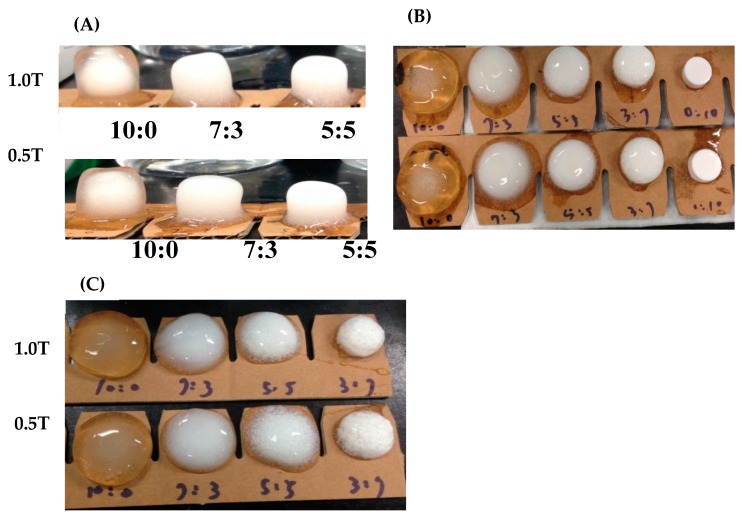
Photographs of different hydrogel tablets after swelling for 24 h in simulated gastric fluid (SGF). (**A**) HPMC 90SH 100K; (**B**) HEC 250HHX; (**C**) PEO7000K.

**Figure 2 pharmaceutics-12-00137-f002:**
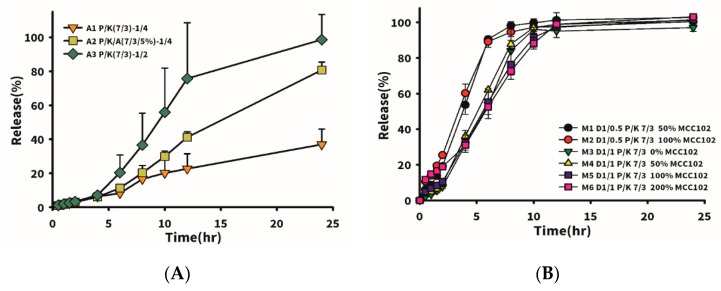
Dissolution profiles of nilotinib in simulated gastric fluid (SGF) from the A1–A3 formulations (**A**) and M1–M6 formulations (**B**).

**Figure 3 pharmaceutics-12-00137-f003:**
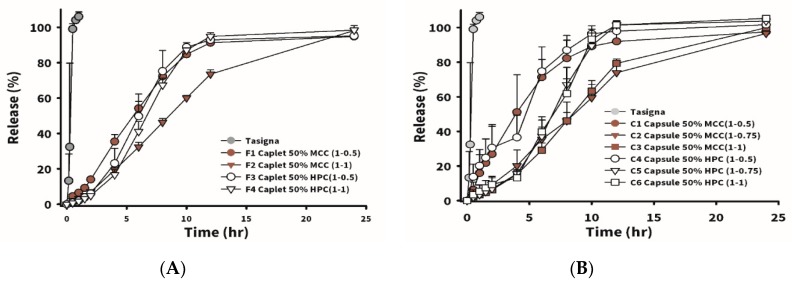
Dissolution profiles of nilotinib in simulated gastric fluid (SGF) from formulations F1–F3 (**A**) and C1–C6 (**B**).

**Figure 4 pharmaceutics-12-00137-f004:**
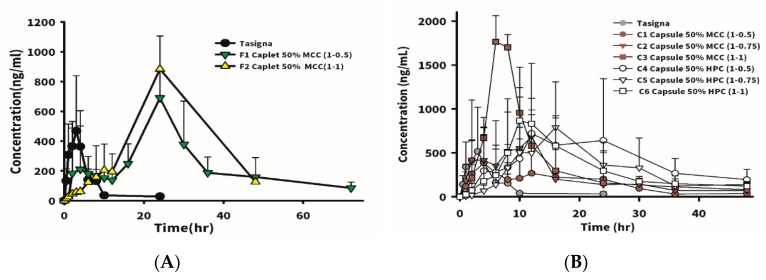
Plasma nilotinib concentration profiles after oral administration of F1 and F2 formulations (**A**) and C1–C6 formulations (mean ± standard deviation, *n* = 2 or 3) (**B**). The commercially available product, Tasigna, was included for comparison.

**Table 1 pharmaceutics-12-00137-t001:** Parameters of the UPLC–MS/MS system for analysis of nilotinib.

Column	BEH C18 (1.7 µm, 2.1 × 50 mm)
Flow rate	0.3 mL/min
Injection volume	5 µL
Column oven	40 °C
Sample oven	4 °C
Mobile phase A (MPA)	0.1% formic acid
Mobile phase B (MPB)	Acetonitrile
Elution condition	0~0.3 min, 90% MPA + 10% MPB
	0.3~1.6 min, 20% MPA + 80% MPB
	2~3 min, 90% MPA + 10% MPB
Capillary voltage	3 kV
Cone voltage	48 V
Desolvation temperature	350 °C
Desolvation gas flow	650 L/h
Collision gas flow	50 L/h

**Table 2 pharmaceutics-12-00137-t002:** Optimized multiple reaction monitoring parameters for nilotinib and imatinib.

Compound	Formula/Mass	Ions	Parent *m*/*z*	Cone Voltage (V)	Daughter	Collision Energy (Units eV)
Nilotinib	C_28_H_22_F_3_N_7_O	1	530.15	56	549.27	28
		2	530.15	56	304.14	26
Imatinib	C_29_H_31_N_7_O	1	494.03	4	246.90	48
		2	494.03	4	394.00	26

**Table 3 pharmaceutics-12-00137-t003:** Swelling and floating abilities of three polymers (HPMC 90SH 100K, HEC 250HHX, and PEO 7000K) with various ratios of Kollidon SR tableted at 0.5 tons of compression force.

Ratios	0		1 h		2 h		5 h		24 h			Suspending		
	D (mm)	T (mm)	D (mm)	T (mm)	D (mm)	T (mm)	D (mm)	T (mm)	D (mm)	T (mm)	Hardness (kPa)	0 h	5 h	24 h
**H/K ratio**														
10:0	12.0	4.63	14.0	7.78	14.6	10.20	15.6	11.84	17.3	20.63	14.1	O	O	O
7:3	12.0	4.40	13.4	7.49	14.2	8.91	15.2	11.15	17.2	16.74	15.7	1 min	O	O
5:5	12.0	4.40	13.6	6.86	14.1	8.40	14.9	10.29	17.3	15.50	20.1	O	O	O
**HEC/K ratio**														
10:0	12.2	4.43	14.6	7.60	18.0	9.15	20.5	10.63	26.8	12.05	8.8	X	--	X
7:3	12.1	4.50	14.3	7.50	16.5	9.20	18.5	10.40	21.1	12.76	9.8	O	--	X
5:5	12.0	4.65	14.3	7.58	16.0	9.45	18.4	10.14	20.5	11.99	17.0	O	--	X
3:7	12.0	4.82	13.8	7.52	15.1	8.49	16.9	9.50	19.0	10.98	23.2	O	--	X
0:10	12.1	5.25	12.1	6.27	12.6	6.57	12.8	7.00	13.7	7.12	21.2	O	--	X
**P/K Ratio**														
10:0	12.0	4.56	15.4	8.26	18.2	9.71	20.7	11.08	27.9	13.15	15.8	O	X	X
7:3	12.1	4.66	15.0	7.98	16.9	9.70	19.5	10.04	24.8	10.39	16.8	O	O	O
5:5	12.0	4.74	14.9	8.25	16.5	8.61	17.8	10.06	22.8	10.82	20.4	O	O	O
3:7	12.0	5.01	13.9	8.18	15.3	8.56	16.1	9.20	17.9	9.74	23.0	O	O	O

K, Kollidon SR; H, HPMC 90SH 100K; HEC, HEC 250HHX; P, PEO7000K; D, diameter; T, thickness.

**Table 4 pharmaceutics-12-00137-t004:** Swelling and floating abilities of three polymers (HPMC 90SH 100K, HEC 250HHX, and PEO7000K) with various ratios of Kollidon SR tableted at 1.0 ton of compression force.

Ratios	0		1 h		2 h		5 h		24 h			Suspending		
	D (mm)	T (mm)	D (mm)	T (mm)	D (mm)	T (mm)	D (mm)	T (mm)	D (mm)	T (mm)	Hardness (kPa)	0 h	5 h	24 h
**H/K ratio**														
10:0	12.0	4.00	13.8	7.98	14.4	8.87	15.3	11.19	18.2	19.73	31.3	X	X	X
7:3	12.0	3.85	13.9	5.93	14.1	7.63	14.9	10.19	16.9	16.17	38.7	X	X	X
5:5	12.0	3.91	13.7	5.94	14.0	7.13	14.6	10.07	16.5	14.90	41.3	X	X	X
**HEC/K ratio**														
10:0	12.0	4.01	14.6	7.4	18.15	9.43	20.4	12.53	28.4	12.96	14.2	X	--	X
7:3	11.9	3.97	14.0	6.94	15.72	10.19	18.8	11.64	23.6	13.04	32.0	X	--	X
5:5	11.8	4.20	13.8	6.4	15.51	8.72	17.3	9.68	21.0	10.73	38.1	X	--	X
3:7	11.9	4.15	13.7	6.62	14.26	8.08	16.3	8.77	17.6	10.89	40.9	X	--	X
0:10	11.9	4.15	12.0	4.79	12.09	5.31	12.4	5.44	12.6	5.69	41.5	X	--	X
**P/K ratio**														
10:0	11.9	3.92	15.9	7.66	18.2	9.17	20.9	10.46	28.8	12.05	36.7	X	X	X
7:3	11.9	4.03	15.3	6.98	17.0	8.83	18.9	10.37	23.1	11.73	40.5	X	X	X
5:5	11.9	4.01	14.8	6.66	16.2	7.92	18.0	9.72	18.5	11.42	42.2	X	X	X
3:7	11.9	4.08	14.1	6.26	15.0	8.28	16.1	9.41	15.7	11.96	42.1	X	X	X

K, Kollidon SR; H, HPMC 90SH 100K; HEC, HEC 250HHX; P, PEO7000K; D, diameter; T, thickness.

**Table 5 pharmaceutics-12-00137-t005:** Composition of formulations A1–A3 and M1–M6.

Formulations	A1	A2 *	A3	M1	M2	M3	M4	M5	M6
Drug (mg)	100	100	100	100	100	100	100	100	100
P/K 7/3	400	400	200	50	50	100	100	100	100
MCC PH102	--	--	--	25	50	0	50	100	200
Tablet (mg)	500	500	300	175	200	200	250	300	400
Hardness (kPa)	5.2	4.5	2.8	1.36	2.00	1.63	2.30	3.03	3.90
Floating	No	Yes	No	1 h		1 h		2 h	

* Mixed with 5% Aerosil 200; P, PEO7000K; K, Kollidon SR.

**Table 6 pharmaceutics-12-00137-t006:** Composition of formulations F1–F4 and C1–C6.

Formulations	F1	F2	F3	F4	C1	C2	C3	C4	C5	C6
Drug (mg)	150	150	150	150	150	150	150	150	150	150
P/K 7/3	75	150	75	150	75	112.5	150	75	112.5	150
MCC PH102	37.5	75	--	--	37.5	56.3	75	--	--	--
HPC_sslSPF_	--	--	37.5	75	--	--	--	37.5	56.3	75
Tablet (mg)	262.5	375	262.5	375	262.5	318.8	375	262.5	318.8	375
Hardness (kPa)	4.37	8.47	5.2	13.56	--	--	--	--	--	--
Floating	No	No	No	No	Yes	Yes	Yes	Yes	Yes	Yes

**Table 7 pharmaceutics-12-00137-t007:** Pharmacokinetics parameters of different formulations of nilotinib in rabbits (mean ± standard deviation, *n* = 2 or 3).

Formulations	C_max_ (ng/mL)	T_max_ (h)	T_1/2_ (h)	AUC_0–72_ (h·ng/mL)	V (L)	Cl (L/h)	BA (folds)
Tasigna	614 ± 363	2.7 ± 1.5	6.7 ± 3.17	2703 ± 1219			1.0
F1	795 ± 311	15.5 ± 10.1	23.53 ± 10.17	16,728 ± 6203			6.0
F2	869	24	NA	19,974			7.4
C1	874 ± 267	9.7 ± 7.9	25.42 ± 19.59	8069 ± 3326	709 ± 336	23.95 ± 9.72	2.65
C2	1353 ± 115	7.0 ± 7.1	21.72 ± 2.98	14,373 ± 7700	395 ± 256	12.18 ± 6.52	4.73
C3	1889 ± 121	7.0 ± 1.4	42.3 ± 8.08	22,870 ± 13,285	455 ± 188	7.89 ± 4.5	7.52
C4	1046 ± 440	15.0 ± 6.0	24.56 ± 7.75	25,500 ± 11,787	276 ± 217	7.1 ± 3.5	8.39
C5	964 ± 214	20.7 ± 8.1	18.81 ± 2.75	18,091 ± 5033	240 ± 83	8.67 ± 2.08	5.95
C6	1053 ± 294	12.0 ± 3.5	56.58 ± 22.69	23,870 ± 186	550 ± 295	6.8 ± 2.2	7.85

C_max_, maximum serum concentration; T_max_, time to reach C_max_; T_1/2_, drug half-life; AUC area under the curve; Cl, clearance; BA, bioavailability.

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
