# Peer review of "Preparation and Characterization of a Novel Swellable and Floating Gastroretentive Drug Delivery System (sfGRDDS) for Enhanced Oral Bioavailability of Nilotinib"

_pharmaceutics, 2020, doi:10.3390/pharmaceutics12020137_

Round 1

Reviewer 1 Report

This is an interesting manuscript and it appears to be acceptable after some amendments.

(1) The introduction is too long. It should be reduced by 50%at least.

(2) Are rabbits good model for humans? The size of pylorus maybe different from human. So, a gastroretentive formulation for rabbits may not be applicable to humans. 

(3) If the data is obtained from the same individual rabbits, is Repeated Measures ANOVA more appropriate? 

(4) What is the fate of the dosage form? What is its residence time in the body?

(5) Any information about the safety of the excipients?

Author Response

Reviewer 1:

(1) The introduction is too long. It should be reduced by 50%at least.

ANS: The introduction is just to give background information from various clinical studies with different dose about that the retention of Nilotinib in the stomach could be an influencing factor determining its oral bioavailability with respective to its pH-dependent solubility. Therefore, we sincerely hope that the introduction could be kept in the present form.

(2) Are rabbits good model for humans? The size of pylorus maybe different from human. So, a gastroretentive formulation for rabbits may not be applicable to humans. 

ANS: The solubility of basic drugs, such as nilotinib, would be expectable to be pH-dependent. It is soluble in acidic pH of stomach whereas it is insoluble in the pH range of small intestine. Since that, the pH value along the GI tract it exposed to after oral administration of nilotinib would be an important factor influencing its bioavailability. Therefore, rabbits selected for this study was because it was indicated in the reference provided by editor that both pH values in stomach and intestine are similar to that in human. It was though that rabbits would be the best choice for the study of the oral bioavailability for a drug of Nilotinib with a pH sensitive solubility with respective to the pH variation along GI tract.

(3) If the data is obtained from the same individual rabbits, is Repeated Measures ANOVA more appropriate? 

ANS: Although the data might be obtained from the same rabbits, a washing out period of longer than one week has been implemented between each study. Therefore, It could not be recognized as a repeated measure design.

(4) What is the fate of the dosage form? What is its residence time in the body?

ANS: Except Kollidon SR, HEC250HHX, HPMC 90SH 100K and PEO7000K all are soluble in the aqueous solution. Based on the in vitro study, it is expectable that all excipients used in this study either dissolve or suspend in the GI fluid 12 to 24 hr after oral administration. Its residence time in stomach might be longer than 12 hr.

(5) Any information about the safety of the excipients?

ANS: All excipients used in this study are pharmaceutical grade. There is no safety issue.

Reviewer 2 Report

The current manuscript describes novel DDS of nilotinib to increase the bioavailability and pharmacokinetic evaluations. The manuscript (especially long introduction section) suffers from numerous flaws, making it difficult to read and evaluate. The experimental methods have some problems as follows.

Abstract and Introduction; the authors describe the toxicity of the drug many times such as “thereby increasing compliance and minimizing side effects.” However, in the current study, the toxicodynamic studies was not performed. Moreover, I cannot understand why the side effects are minimum when the bioavailability of the drug is increased? The abstract section was too long to understand the aim of this study. The major revision should be needed. Methods; in PK study, the sample size is too small (n=3) in order to evaluate the pharmacokinetic characteristics using animal. How did the authors determine the sample size in the current study? Regarding the above comments, in the table 7 and figure 4, the standard deviations could not be determined in the case of n=2. This is the serious mistakes in the statistical analysis. Why was the results n=2 when the drugs were administrated to three animals? The side effects were observed and lead the death? The blood sampling were performed until 72h after dosing. However, in the figure 4, the observed data at 48 and 72h were not demonstrated. Figure 2 and 3; the observed data at 24h were not shown. The mean and standard deviations at all sampling points should be demonstrated. In the current study, the amount of drug content was not evaluated in the preparations. The containing dose of150mg nilotinib lead the wrong pharmacokinetic parameters such as BA. This is an important issue for pharmaceutical study. The loss of the drug in making process of preparations for each preparation should be evaluated. The stability of the drug in the preparations was not evaluated and should be investigated. Please describe the version of WinNonlin software.

Author Response

Reviewer 2

Abstract and Introduction; the authors describe the toxicity of the drug many times such as “thereby increasing compliance and minimizing side effects.” However, in the current study, the toxicodynamic studies was not performed. Moreover, I cannot understand why the side effects are minimum when the bioavailability of the drug is increased?

ANS: Regarding the minimization of side effects, it was based on the Cmax and AUC0-24 that could be accomplished after the administration of resepctive dosage regimen. With giving the same single dose as Tasigna, when the oral administration of GRDDS formulation resulting in a similar even a less Cmax but greater AUC0-24 would indicate that one same single dose of GRDDS formulation as Tasigna could achieve its minimal effective concentration and maintain at this concentration with a longer period of time. Since Tasigna was taken two times a day at a dosing amount of 300-400 mg, it was expectable that the resulting Cmax would be 2-3 folds higher than that achieved by single dose of 150 mg Tasigna. This demonstrated it would be possible to decrease the side effect in turn of less Cmax concentration achieved by the administration of one single dose of 150 mg GRDDS.

The abstract section was too long to understand the aim of this study. The major revision should be needed.

ANS: There is limited word number for the abstract section set by the Journal. The word number of the abstract section conforms to this limit.

Methods; in PK study, the sample size is too small (n=3) in order to evaluate the pharmacokinetic characteristics using animal. How did the authors determine the sample size in the current study? Regarding the above comments, in the table 7 and figure 4, the standard deviations could not be determined in the case of n=2. This is the serious mistakes in the statistical analysis. Why was the results n=2 when the drugs were administrated to three animals? The side effects were observed and lead the death?

ANS: Since this was a preliminary study to screen novel GRDDS formulations that might be able to enhance the oral bioavailability of Nilotinib, three rabbits selected in PK study was just to have number of animals that could be evaluated statistically to prove the concept that GRDDS could enhance the oral bioavailability for a basic drug such as Nilotinib. There was only formulation F2 having evaluable data of n=2 due to one of three animal data being loss as a result of malfunction of instrument during the analysis of plasma samples. Therefore, only mean values were present in Table 7.

The blood sampling were performed until 72h after dosing. However, in the figure 4, the observed data at 48 and 72h were not demonstrated. Figure 2 and 3; the observed data at 24h were not shown. The mean and standard deviations at all sampling points should be demonstrated.

ANS: All blood samples were withdrawn until 72 h after dosing. Data for 48 and 72 h in Figure 4 and data for 24 in Figure 2 and 3 was due to this portion of graphs being cut off.

In the current study, the amount of drug content was not evaluated in the preparations. The containing dose of 150mg nilotinib lead the wrong pharmacokinetic parameters such as BA. This is an important issue for pharmaceutical study. The loss of the drug in making process of preparations for each preparation should be evaluated.

ANS: All tablets and capsules with an amount equivalent to 150 mg of Nilotinib were individually filled in the die for tableting or directly filled into each capsule. By this way, the exact amount of 150 mg Nilotinib was expectable to be loaded in each tablet, caplet, or capsule with no loss of drug.

The stability of the drug in the preparations was not evaluated and should be investigated.. 

ANS: All excipients used in this study are pharmaceutically acceptable and are biocompatible with Nilotinib. A full stability study for clinical batch would be conducted to examine the influence of those excipients on the stability of Nilotinib.

Please describe the version of WinNonlin software

ANS: the version of WinNonlin software is WinNonlin 6.3 software (Pharsight®, Princeton, NJ, USA)

Round 2

Reviewer 1 Report

The manuscript has been improved.

Author Response

The manuscript has been improved.

    ANS: Thank you.  

Reviewer 2 Report

In my opinion, the revision version was not fully improved and the answer to my comments are not fully accepted.

As the Reviewer 1 mentioned, the Introduction section was too long to understand the aim of this study, and should be correct in order to concise for readers.

Regarding to the concern about the figure 2-4, “Data for 48 and 72 h in Figure 4 and data for 24 in Figure 2 and 3 was due to this portion of graphs being cut off.” could not accepted. In the figure, the mean value and SD data at 24h, 48, or 72h was not shown whereas there was the link line between 12 h and 24h (36 h and 48h, or 48 h and 72 h). What is the cut off? Why should the author apply the cut off?

If the author did not performed the stability test in the current study, this should be described as the limitation.

Round 3

Reviewer 2 Report

The revised manuscript has been improved.